# Cyanobacteria Secondary Metabolites as Biotechnological Ingredients in Natural Anti-Aging Cosmetics: Potential to Overcome Hyperpigmentation, Loss of Skin Density and UV Radiation-Deleterious Effects

**DOI:** 10.3390/md20030183

**Published:** 2022-03-01

**Authors:** Rita Favas, Janaína Morone, Rosário Martins, Vitor Vasconcelos, Graciliana Lopes

**Affiliations:** 1CIIMAR/CIMAR, Interdisciplinary Centre of Marine and Environmental Research, Novo Edifício do Terminal de Cruzeiros do Porto de Leixões, Avenida General Norton de Matos, S/N, 4450-208 Matosinhos, Portugal; rita.favas@hotmail.com (R.F.); janabavini@ciimar.up.pt (J.M.); mrm@ess.ipp.pt (R.M.); vmvascon@fc.up.pt (V.V.); 2FCUP, Department of Biology, Faculty of Sciences, University of Porto, Rua do Campo Alegre, 4169-007 Porto, Portugal; 3CISA, Health and Environment Research Centre, School of Health, Polytechnic Institute of Porto, Rua Dr. António Bernardino de Almeida, 400, 4200-072 Porto, Portugal

**Keywords:** melanin, UV-blocker, elastase, hyaluronidase, tyrosinase, oxidative-stress, beta-carotene, phycocyanin, phycoerythrin, allophycocyanin

## Abstract

The loss of density and elasticity, the appearance of wrinkles and hyperpigmentation are among the first noticeable signs of skin aging. Beyond UV radiation and oxidative stress, matrix metalloproteinases (MMPs) assume a preponderant role in the process, since their deregulation results in the degradation of most extracellular matrix components. In this survey, four cyanobacteria strains were explored for their capacity to produce secondary metabolites with biotechnological potential for use in anti-aging formulations. *Leptolyngbya boryana* LEGE 15486 and *Cephalothrix lacustris* LEGE 15493 from freshwater ecosystems, and *Leptolyngbya* cf. *ectocarpi* LEGE 11479 and *Nodosilinea nodulosa* LEGE 06104 from marine habitats were sequentially extracted with acetone and water, and extracts were analyzed for their toxicity in cell lines with key roles in the skin context (HaCAT, 3T3L1, and hCMEC). The non-toxic extracts were chemically characterized in terms of proteins, carotenoids, phenols, and chlorophyll *a*, and their anti-aging potential was explored through their ability to scavenge the physiological free radical superoxide anion radical (O_2_^•−^), to reduce the activity of the MMPs elastase and hyaluronidase, to inhibit tyrosinase and thus avoid melanin production, and to block UV-B radiation (sun protection factor, SPF). *Leptolyngbya* species stood out for anti-aging purposes: *L. boryana* LEGE 15486 presented a remarkable SPF of 19 (at 200 µg/mL), being among the best species regarding O_2_^•−^ scavenging, (IC_50_ = 99.50 µg/mL) and also being able to inhibit tyrosinase (IC_25_ = 784 µg/mL), proving to be promising against UV-induced skin-aging; *L. ectocarpi* LEGE 11479 was more efficient in inhibiting MMPs (hyaluronidase, IC_50_ = 863 µg/mL; elastase, IC_50_ = 391 µg/mL), thus being the choice to retard dermal density loss. Principal component analysis (PCA) of the data allowed the grouping of extracts into three groups, according to their chemical composition; the correlation of carotenoids and chlorophyll *a* with MMPs activity (*p* < 0.01), O_2_^•−^ scavenging with phenolic compounds (*p* < 0.01), and phycocyanin and allophycocyanin with SPF, pointing to these compounds in particular as responsible for UV-B blockage. This original survey explores, for the first time, the biotechnological potential of these cyanobacteria strains in the field of skin aging, demonstrating the promising, innovative, and multifactorial nature of these microorganisms.

## 1. Introduction

Skin is a large and surprisingly complex human organ with a primordial barrier function of protecting internal organs from harmful stressors, such as chemicals, pathogens, cold, heat, and ultraviolet radiation (UVR). However, the importance of the skin goes much further, encompassing an undeniable socio-cultural role. In fact, skin appearance and shape are of crucial importance to an individual’s self-esteem, its care and beautification being part of the daily routine since ancient times [1].

The delay of the skin-aging process has been a main societal demand. This slow and complex process is induced by endogenous and exogenous factors that predispose skin to a progressive structural and functional degeneration that, beyond affecting its aesthetic appearance, leaves it prone to the development of a wide variety of diseases. Among exogenous factors, UVR is perhaps the most harmful agent to which the skin is most exposed. Excessive UVR exposure increases free radical generation, triggering a cascade of events that affect a wide variety of cell structures and enzymes, resulting in an immediate inflammatory response, and finally culminating in premature skin aging [2]. A particularly affected target in this theme is the extracellular matrix (ECM), a three-dimensional network of elastin and collagen fibers surrounded by the ground substances, such as hyaluronic acid (HA), that act together to maintain skin filling, elasticity, and flexibility [3]. Under a framework of oxidative-stress, elastin and collagen fibers, responsible for the elasticity and resistance of the skin, may lose their structure [4]. This phenomenon, partly due to the deregulated activity of the matrix metalloproteinases (MMPs), such as elastase, collagenase, and hyaluronidase, is at the base of premature skin aging associated with external factors [1]. These alterations affect the epidermal thickness, structure, and appearance, resulting in dryness, enlarged pores, fine lines, and wrinkles [5,6]. Therefore, the search for new, effective, multitarget, and innovative ingredients for cosmetic formulations able to reach the greatest number of key targets in the skin aging process has been the focus of the cosmetics industry in recent decades.

The growing research on natural sources, more specifically those of marine origin, has provided a countless number of new molecules with promising bioactivities, worthy of further exploitation in the field of skin aging [1]. Among them, cyanobacteria stand out due to their capacity to produce bioactive secondary metabolites with unique structures and mechanisms of action. Beyond representing the only group of prokaryotes that can perform oxygenic photosynthesis, similarly to plants, cyanobacteria are self-renewable, have basic nutritional requirements, require minimal cultivation space, and have a low environmental impact, being a sustainable choice as a green source of cosmetic ingredients [7].

Focusing on skin formulations, cyanobacteria produce a wide array of bioactive metabolites, including phenolic compounds, proteins, pigments, and MMPs inhibitors, that cover the main processes comprising the basis of skin aging, namely antioxidation, photoprotection, and the ability to inhibit crucial enzymes of the ECM [8,9,10,11,12], as recently reviewed [1,9]. Regarding antioxidant activity, numerous strains have been reported to have a representative amount of metabolites with radical scavenging capacity, such as carotenoids, e.g., those from the genera *Leptolyngbya*, *Synechocystis,* and *Wollea* [9,13], phycobiliproteins (PBP), such as *Arthospira* spp. and *Spirulina* spp. [13,14], and phenolic compounds, such as *Nostoc commune* [15]. In the field of photoprotection, cyanobacteria also stand out through the production of the well-known UVR-absorbing compounds mycosporine-like amino acids (MAAs) and scytonemin (SCY) [16]. Concerning the potential to inhibit collagenases, elastase, hyaluronidases, and tyrosinase, several examples are reported in the literature, such as mycosporine-2-glycine (M2G), isolated from the cyanobacterium *Aphanothece halophytica* [17], the cyclic depsipeptides tutuilamides A−C, from *Schizothrix* spp. and *Coleofasciculus* spp. [18], a polysaccharide from *Nostochopsis lobatus* MAC0804NAN [19], and the extract Phormiskin Bioprotech G^®^, from *Phormidium persicinum* [20], respectively.

Despite the studies presented above, the number of cyanobacteria strains explored in the field of cosmetics is still very low, considering the potentialities of this resource. With this in mind, this survey explores, for the first time, the biotechnological potential of sequential extracts of different polarities, obtained from four cyanobacteria strains, in the field of skin aging, demonstrating the promising, innovative, and multifactorial nature of these microorganisms.

## 2. Results and Discussion

In the present study, two freshwater and two marine cyanobacteria strains were cultured, harvest, and subjected to a sequential extraction with acetone and water, with a view to their exploitation for cosmetic purposes. The sequential extraction with acetone followed by water was designed to monetize the biomass, in a more sustainable, environmentally friendly, and economically attractive process. In Table 1, the extraction yields are displayed.

The extraction yield was significantly higher with water than with acetone (*p* < 0.05), which has a direct correlation with the affinity of the different compounds to each solvent, their molecular weight, and polarity, as later discussed in this study. It should be noted that *Leptolyngbya boryana* LEGE 15486 was the strain which showed higher extraction yields, what could be economically interesting when considering possible industrial applications.

### 2.1. Extracts Cytotoxicity

In order to select the extracts to proceed for biological activities assessment, an in vitro cytotoxicity assay was performed using different cell lines. Cytotoxicity tests are essential during cosmetics production, since they may predict health risks connected with the use of the extracts as bioactive ingredients. Thus, the cytotoxicity of the extracts was evaluated in three different cell lines of key importance regarding products for skin application: keratinocytes (HaCat), fibroblasts (3T3L1), and endothelial cells (hCMEC). Fibroblasts are among the most important cells regarding skin aging, being responsible for the production of the dermal matrix components essential for the maintenance of skin shape and structure, such as collagen and hyaluronic acid [9,21]. Endothelial cells form a barrier between vessel walls and blood, and were chosen to account for their presence in the dermis which, contrary to the epidermis, is an irrigated layer of the skin [22]. Regarding keratinocytes, they are the key cells of the epidermis, composing about 95% of this layer, but are present in all four layers, and provide structure and defense to the skin [23,24]. None of the extracts presented toxicity for the selected cell lines under the tested concentrations (12.5–200 µg _dry extract_/mL) (Appendix A). In this regard, all the extracts followed to the next step, where they were subjected to chemical characterization and evaluation of their biological activities.

### 2.2. Chemical Profile

The acetone and water extracts were chemically characterized in terms of phenols, total proteins, PBPs, total carotenoids, and chlorophyll *a*, in order to compare the chemical profiles obtained with the different extraction solvents, and to establish a relationship between the chemical composition and the biological activities evaluated.

#### 2.2.1. Total Phenolic Content (TPC)

The total phenolic content (TPC) of both extracts was measured through the Folin–Ciocalteu colorimetric assay. Even considering the inherent limitations of the method, this is a standard assay widely used for the quick determination of total phenols, allowing the comparison of different samples, and consequently predicting their antioxidant potential. Table 2 displays the data (expressed in GAEs) for the total phenolic content of the eight extracts explored in the present study. The highest TPC was found in the acetone extract of *Leptolyngbya* cf. *ectocarpi* LEGE 11479, with 17.59 µg GAE/mg _dry extract_, followed by the water extracts of *Leptolyngbya boryana* LEGE 15486 and *Cephalothrix lacustris* LEGE 15493 (Table 2). Considering the yield of the combined extraction, *Leptolyngbya* cf. *ectocarpi* LEGE 11479 was the richest species, totaling about 26 µg GAE/mg _dry extract_ when considering both extraction solvents.

Overall, the water extracts of the freshwater strains presented higher phenolic content than those from marine environments, while the opposite behavior was observed regarding acetone extracts. This observation was also noted for the two strains of the same genera, what leads to the assumption that, beyond species-specific characteristics, marine strains are more likely to produce phenolic compounds of lower polarity, given their higher prevalence in less polar solvents. Phenols have a polar and a nonpolar component in their molecules, possessing a solubility preference to solvents of intermediate polarities such as alcohols and acetone, rather than water. However, the solubility of phenols in different solvents cannot be based only on their polarities, since other parameters such as temperature and pH can have a great influence on their solubility, and thus justify their expressive presence in water extracts [25].

Previous work from our research group also explored TPC in different cyanobacteria extracts. Morone and co-workers [26] used the same methodology for TPC quantification, and found that *Nodosilinea nodulosa* LEGE 06102 had a value of 1.23 mg GAE/g _dry biomass_, which was above the values obtained herein (0.59 mg GAE/g _dry biomass_, converted according to the extraction yield). However, the authors used 70% ethanol as an extraction solvent, which certainly led to the differences observed. For all the strains analyzed by the authors, the highest value was found in *Synechocystis salina* LEGE 06099 (2.45 mg GAE/g). In another study, with *Nodosilinea antarctica* LEGE 13457, the TPC content was 19.23 µg GAE/mg (acetone extract) and no phenols were detected in *Leptolyngbya*-like sp. LEGE 13412, which demonstrated the highest variability among similar species. They also reported data for other strains, with *Cyanobium gracile* LEGE 12431 presenting the highest phenol content, with 22.01 µg GAE/mg of acetone extract [27]. Trabelsi and his team [28] reported that the thermophilic cyanobacterium *Leptolyngbya* sp. possessed 139 mg GAE/g, this being the among the highest values reported in the literature, and attributed by the authors to the high temperature of the cyanobacterial habitat: to avoid oxidative stress induced by high temperatures, cyanobacteria produce higher amounts of antioxidant compounds such as phenols and flavonoids. Another research group [29] also reported a TPC of 6.24 mg GAE/g for *Leptolyngbya* sp. KC45, sampled from a location with temperatures of approximately 40–45 °C.

Despite the differences inherent to the effects of the extraction solvents, and to the cyanobacterial cultivation conditions, such as light conditions, culture medium nutrients, and cell density, among others, the TPC values obtained herein are within the same order of magnitude as those reported in the literature. Moreover, *Leptolyngbya boryana* LEGE 15486 can be pointed out as an interesting species regarding phenolic compounds (3.08 mg GAE/g dry biomass), when compared to the widely known *Spirulina* spp. (1.78 mg GAE/g) [30].

#### 2.2.2. Proteins

Although proteins are generally undervalued, they compose a large fraction of the cyanobacterial biomass [31], with reported antioxidant and immunostimulant properties, as well as the ability to confer moisture retention to the skin, which is essential to prevent skin aging [32]. With this in mind, the extracts explored in the present study were characterized for their protein content, through a general approach focusing on total proteins (acetone and water extracts) (Table 3), and a targeted approach focusing on PBPs (water extracts) (Table 4).

Regarding total proteins, it can be concluded that the aqueous extraction resulted in higher values than the acetone extraction, and also, that marine strains were poorer than those from freshwater environments. In terms of dry biomass, *Leptolyngbya boryana* LEGE 15486 presented the highest content (69.22 mg BSA/g _dry biomass_), and *Nodosilinea nodulosa* LEGE 06104 the lowest one (15.23 BSA/g _dry biomass_). Among other factors, these differences may be due to the different amount of structural proteins presented by different cyanobacteria strains [32].

A practical example of the use of proteins in cosmetics concerns PBSs, which are naturally present in cyanobacteria. One of cyanobacteria’s defensive mechanisms is the capacity of these macromolecules to absorb light energy without producing reactive oxygen species (ROS), which is possible due to the changes in their content and ratio in phycobilisomes [33]. PBPs are water-soluble proteins that are associated with phycobilins, and divided into three groups according to their structure and light absorption spectra: phycocyanin (PC, 610–625 nm), phycoerythrin (PE, 490–570 nm), and allophycocyanin (APC, 650–660 nm). As stated earlier, their interest as cosmetic ingredients is mainly due to their recognized antioxidant potential, thanks to their structural resemblance to bilirubin, which eliminates oxygen derivatives. PC is the most common PBP in cyanobacteria, with interesting antioxidant and radical scavenging properties, as well as the capacity to inhibit cell proliferation [13,14].

Regarding PC, *Leptolyngbya boryana* LEGE 15486 possessed the highest content (154.07 µg/mg _dry extract_), followed by *Cephalothrix lacustris* LEGE 15493 (115.03 µg/mg _dry extract_), which is in accordance with the strong blue color observed in the aqueous extracts of these species. On the other hand, *Leptolyngbya* cf. *ectocarpi* LEGE 11479, with an intense pink/purple coloration in the biomass and in the aqueous extract, was the richest in PE (138.73 µg/mg _dry extract_). *Nodosilinea nodulosa* LEGE 06104, with the lightest blue extract, presented visibly lower values of both PBPs (Table 4).

When comparing strains in terms of PBPs, it is extremely important to consider the culture conditions because differences in light, nitrogen, temperature, pH, carbon, and salinity can drastically influence their production [14]. Pumas and co-workers [29] evaluated the PBP content of the thermotolerant cyanobacteria *Leptolyngbya* sp. KC45, and found a PE content of almost 100 mg/g, followed by approximately 40 and 43 mg/g of APC and PC, respectively, which is in line with the results obtained herein for *Leptolyngbya* cf. *ectocarpi* LEGE 11479 (Table 4). The work performed by Pagels and co-workers also explored the content of PBPs in sequential extracts obtained with different solvents. Although exploring a different cyanobacteria strain (*Cyanobium* sp.), the authors found a total PBP content close to 200 mg/g in a sequential extraction using the same solvents as those used herein, which is in accordance with the values obtained by us [34]. When comparing with the widely known *Spirulina* sp., with reported total PBPs of about 19 mg/g _dry biomass_ [30], it is possible to consider our strains of economic interest. In the same study, the authors also reported values of 127.01 mg/g _dry biomass_ of total PBPs for *Lyngbya* sp. In another study, data were presented from 18 strains, where the highest amount was found in *Anabaena* NCCU-9, with a value of 91.54 mg/g _dry biomass_ [35].

#### 2.2.3. Carotenoids and Chlorophyll *a*

Carotenoids are natural fat-soluble isoprenoids with a wide array of colorations, varying from yellow to red [13]. These compounds have gained increased attention for their association with a decreased risk of several degenerative disorders, and for their recognized antioxidant activity, partly related to the C=C chemical double bonds present in their molecules. Besides these functions, carotenoids act as UV filters by reducing light exposure [36]. Some cyanobacteria such as *Wollea vaginicola*, *Leptolyngbya foveolarum*, and *Synechocystis salina* LEGE 06099 have been highlighted for their significant carotenoid content, enhancing their interest in the pharmacological and cosmetic fields [9,13].

The total carotenoid and chlorophyll *a* contents of the cyanobacteria acetone extracts explored herein are displayed in Table 5. The carotenoid amount ranged from 89 to 159 µg/mg _dry extract_, *Leptolyngbya boryana* LEGE 15486 being the richest strain. Regarding chlorophyll *a*, all the strains presented values greater than 100 µg/mg _dry extract_, with *Cephalothrix lacustris* LEGE 15493 standing out.

Other cyanobacteria have also been the subject of pigment evaluation. For instance, Lopes and her team [27] reported 63.9 µg/mg _dry extract_ of total carotenoids and 417.6 µg/mg _dry extract_ of total chlorophylls for *Nodosilinea antarctica* LEGE 13457 acetone extract. The authors also provided data for *Leptolyngbya*-like sp. LEGE 13412 (33.6 µg/mg _dry extract_), and in both cases, the carotenoid amount was below the values reported herein. It is worth mentioning that, in the work referred to above, carotenoids were profiled by HPLC and thus, despite species-specific characteristics, the methodology justifies the differences obtained. Regarding chlorophylls, the amount reported by the authors was greater than that obtained herein, which is easily justified by the fact that they present the value of total chlorophylls, whereas the work herein focuses solely on chlorophyll a and its derivatives.

Another study from our research group reported a total of 0.37 mg/g _dry biomass_ of carotenoids for the ethanol extract of *Nodosilinea nodulosa* LEGE 06102 [26], which was below the values obtained herein (0.75 mg/g _dry biomass_, converted based on the extraction yield). This difference is certainly correlated with the solvent used in the extraction since, due to their chemical characteristics, carotenoids have a higher affinity for acetone. Generally speaking, the results obtained herein are within the same order of magnitude as those obtained by other authors, what may be explained by the very similar culture conditions and methodologies used.

### 2.3. Biological Activities

#### 2.3.1. Radical Scavenging Activity

The superoxide anion radical is a physiological free radical with extreme importance for human body. When there is an overproduction of this ROS during aerobic respiration, or when the endogenous detoxification mechanisms fail or are insufficient, there is an increased risk of oxidative damage with serious deleterious effects. In this sense, finding mechanisms to inhibit the deleterious effects of O_2_^•−^ is of key importance, not only in the field of cosmetics, but also considering the amelioration and prevention of a wide array of diseases. The O_2_^•−^ scavenging behavior of cyanobacteria extracts is displayed in Figure 1, and the IC values are summarized in Table 6.

The aqueous extracts were significantly more effective in O_2_^•−^ scavenging than the acetone extracts (Table 6) and presented a dose-dependent activity (Figure 1), with the freshwater strains standing out from the marine strains. *Cephalothrix lacustris* LEGE 15493 was the most effective strain, presenting the lowest IC_50_ value (65.5 µg dry extract/mL, *p* < 0.05), followed by *Leptolyngbya boryana* LEGE 15486 and *Nodosilinea nodulosa* LEGE 06104. *Leptolyngbya* cf. *ectocarpi* LEGE 11479 was the only strain which did not reach the IC_50_ for the aqueous extract (Table 6). On the other hand, the acetone extract of this strain was the most effective in O_2_^•−^ scavenging.

A comparison of the antioxidant capacities among different cyanobacteria is challenging due to the different methods applied. Morone and co-workers evaluated the radical scavenging ability of ethanol (70% *v/v*) extracts of different cyanobacteria strains [26], the lowest IC_50_ value being 822.70 µg/mL for *Phormidium* sp. LEGE 05292, while no activity was detected for *Nodosilinea nodulosa* LEGE 06102. Lopes and co-workers reported O_2_^•−^ scavenging activity for *Nodosilinea* (*Leptolynbbya*) *antarctica* LEGE13457 and *Cuspidothrix issatschenkoi* LEGE 03282 acetone extracts, with IC_25_ values of 319 and 286 µg/mL, and no activity for *Leptolynbbya*-like sp. LEGE 13412 [27]. The authors also reported a higher effectiveness in the acetone extracts when compared to ethanol extracts. Another study conducted by Amaro and co-workers revealed IC_50_ values of 1394 and 826 µg/mL for *Gloeothece* sp. and *Scenedesmus obliquus* (M2-1), respectively [37]. The results obtained herein for acetone extracts seem less promising than those previously reported, even though they are within the same order of magnitude. On the other hand, the aqueous extracts revealed an enormous potential worthy of further exploitation regarding cosmetic applications in the field of skin aging.

#### 2.3.2. Enzymes Inhibition

MMPs are a family of extracellular zinc-dependent enzymes, whose main function is to remodel and degrade the ECM [38], a gel-like material essential to hold cells together and provide a pathway for nutrients and oxygen [39]. Alterations in the ECM components, such as collagen and elastin, induced by MMPs are the basis of skin damage and wrinkle formation [40]. Together with these enzymes, linked with skin structure and wrinkle formation, another assumes a crucial role in the aging process due to its activity in melanogenesis: tyrosinase. Among other factors, UVR exposure causes an accumulation of an abnormal amount of melanin, due to increased ROS production. These reactive species affect the activity of melanocytes, which increases the conversion of tyrosine into melanin by oxidation, leading to hyperpigmentation and irregular skin patches [41].

In the search for natural alternatives to commercial anti-aging ingredients, focusing the enzymes mentioned above, the activity of cyanobacteria extracts was explored (Figure 2). Regarding HAase, only three strains were able to inhibit this enzyme, acting in a dose-dependent manner: the aqueous extract of *Leptolyngbya* cf. *ectocarpi* LEGE 11479 (IC_50_ = 863 µg/mL), and the acetone extracts of *Cephalothrix lacustris* LEGE 15493 and *Nodosilinea nodulosa* LEGE 06104, the latter two only reaching IC_25_ (832 and 995 µg/mL, respectively). Even though these values seem high, their range of activity was similar to that of the reference drug di-sodium cromoglicate (DSCG) (IC_50_ = 1105 µg/mL). Moreover, it is worth mentioning that, for the highest concentration tested (1 mg/mL), *Leptolyngbya* cf. *ectocarpi* LEGE 11479 inhibited this enzyme in 80% (Figure 2), which makes this extract promising as a cosmetic ingredient.

Few studies have reported on the potential effect of cyanobacteria compounds and extracts on hyaluronidase activity, and any comparison on extracts’ biological potential should take into account that cyanobacterial metabolism suffers significant variation depending on the cultivation conditions, influencing extracts’ chemical composition. Morone and co-workers [26] evaluated the effect of ethanol extracts of *Tychonema* sp. LEGE 07196 and *Cyanobium* sp. LEGE 07175 on hyaluronidase, and found a stronger inhibitory activity, with IC_50_ values of 182.74 and 208.36 µg/mL, respectively. The activity of ethanol extracts has also been reported for an insoluble fraction of *Spirulina platensis*, with an IC_50_ of 150 µg/mL [42]. Another study, conducted by Yamaguchi and Koketsu [19], showed that *Nostochopsis lobatus* MAC0804NAN produced a large amount of polysaccharides with a high inhibitory effect (IC_50_ = 7.18 µg/mL). It was also reported that an *Arthrospira*-derived peptide may be involved in hyaluronidase inhibition [43]. Together, these results support the potential of cyanobacteria compounds and extracts as anti-aging ingredients for cosmeceutical applications.

Considering elastase, only acetone extracts presented interesting bioactivity. *Leptolyngbya* cf. *ectocarpi* LEGE 11479 was again the most active (Figure 2), being the only strain to reach the IC_50_, with a value of 391 µg/mL. *Nodosilinea nodulosa* LEGE 06104, *Cephalothrix lacustris* LEGE 15493, and *Leptolyngbya boryana* LEGE 15486 only reached IC_25_, with values of 126, 86, and 99 µg/mL, respectively. As for hyaluronidase, marine strains have shown to be the most promising.

To the best of our knowledge, there are no previous reports on elastase inhibitory activity for the strains explored herein. Regarding other strains, it was discovered that *Nostoc minutum* produced microviridins-type peptides and nostopeptins, with IC_50_ = 1.3 and 11.0 µg/mL [44,45]. Microviridins B and C obtained from *Microcystis aeruginosa* also inhibited elastase effectively, with IC_50_ values of 0.044 and 0.084 µg/mL [46]. Most of the available data regarding elastase inhibition focus on isolated compounds, so comparisons with the extracts from our strains are difficult to make.

Uneven skin pigmentation, associated with both aging and UV exposure, remains a major concern of aging populations and cosmetic industries. The majority of the available studies are trifling and use mushroom tyrosinase as an enzymatic model, making it difficult to translate the results to the human environment, nevertheless, this enzyme has a high similarity and homology with human tyrosinase [47]. Thus, the same enzymatic model was used herein to explore the potential of cyanobacteria extracts in tyrosinase inhibition. As for elastase, only acetone extracts were able to inhibit tyrosinase. *Nodosilinea nodulosa* LEGE 06104 was the most effective (Figure 2), being the only strain to reach the IC_50_ (989.26 ± 4.3 µg/mL). Below that was *Leptolyngbya boryana* LEGE 15486, with IC_25_ = 784. 78 ± 4.33 µg/mL, and lastly *Leptolyngbya* cf. *ectocarpi* LEGE 11479, with the most promising results being found for the marine strains.

Morone and co-workers [26] previously explored the activity of cyanobacteria ethanolic extracts in the same model, but found no activity. One interesting aspect of this is that one of the studied strains was *Nodosilinea nodulosa* LEGE 06102, which showed the best results herein, once again emphasizing the importance of the extraction solvents in obtaining targeted bioactive extracts. As for the other enzymes explored, surveys focusing cyanobacteria are also scarce for tyrosinase. In work conducted by Yabuta and his team [48], it was reported that the hot water extract of *Nostochopsis* spp. significantly inhibited the tyrosinase activity (IC_50_ = 250 µg/mL). This is a very interesting result, considering that it results from an aqueous extract, for which no activity was found in the present study. The authors attribute the result to the low molecular weight compounds released from PBPs by heat treatment, namely a biline moiety that acts as a potent peroxyl radical scavenger. Another study evaluated the inhibitory activity of *Arthrospira platensis* ethanol (IC_50_ = 14,000 µg/mL) and water (IC_50_ = 72,000 µg/mL) extracts, where the values were attributed to the presence of phenolic compounds such as ferulic and caffeic acids in the ethanol extract [49].

#### 2.3.3. UV Protection

Even though a growing number of cosmetic companies incorporate sun blockers into their products, it is still difficult to convince consumers about the benefits of their daily use to slow down premature skin aging. If on the one hand, the daily application of sun blockers is a little ingrained habit, on the other hand, there is a certain fear in the use of synthetic substances, due to their unwelcome associated risks [50]. Following on from this, research on natural photoprotectors has considerably increased in the last few years, considering their potential for biodegradability and lower toxicity, making them more beneficial to humans and the environment. In order to explore cyanobacteria extracts in this field, their capacity to act as sunscreens was evaluated in vitro for UVR-B, since it is the most harmful radiation. The results found for cyanobacteria aqueous and acetone extracts are presented in Table 7.

Regarding acetone extracts, the most promising value (19.2) was found for *Leptolyngbya boryana* LEGE 15486, followed by *Leptolyngbya* cf. *ectocarpi* LEGE 11479 (10.7), both at the lowest concentration tested, 200 µg/mL. *Nodosilinea nodulosa* LEGE 06104 was the least promising among acetone extracts. In aqueous extracts, the most promising results were obtained for the highest concentration tested, with *Leptolyngbya boryana* LEGE 15486 and *Cephalothrix lacustris* LEGE 15493 standing out with in vitro SPF values of 17.1 and 14.9, respectively (Table 7). Discussing previous studies in this field, Hossain and his team [51] reported that the SPF value for *Cephalothrix*
*komarekiana* extract was 2.37. Another group found that the SPF of a methanol extract of *Aphanizomenon flos-aquae* was 4 [52]. However, the information about the concentration of the extract used by the authors was not available, making possible comparisons difficult.

The number of cyanobacteria strains explored in the field of cosmetics, especially regarding SPF, is very scarce, considering the potentialities of these resources. The results presented herein demonstrate that the species under study may be good options as biological photoprotectors, and possibly act as boosters to other sunscreens currently marketed, allowing a reduction in the concentration of synthetic sunscreens in the formulas. Therefore, it is crucial to increase research on these organisms, especially their bioactive extracts, that, being of significantly lower cost, higher yield, and faster obtainment than isolated compounds are more environmentally friendly and economically more attractive.

### 2.4. Discrimination of Cyanobacteria Extracts by PCA Analysis

The search for bioactive compounds from natural sources, with the potential for use as ingredients in the field of cosmetics has grown in recent years. In addition to being potentially less toxic and completely biodegradable, cyanobacteria-derived compounds are available from renewable sources and can be obtained at low cost in a controlled environment. The classification of bioactive extracts according to their chemical composition and biological activities can be valuable to point out potential relationships. In this regard, principal component analysis (PCA) was applied, considering the chemical composition of cyanobacteria extracts and the bioactivity of the highest concentration tested in each assay (Figure 3). As can be observed, 72.99% of the variability could be explained by the first two dimensions: PC1 accounted for 50.41% of the variance, while PC2 was responsible for 22.58%. Three groups were distinguished (Figure 3A): G1, involving the *Leptolyngbya* cf. *ectocarpi* LEGE 11479 water extract; G2, involving the *Cephalothrix lacustris* LEGE 15493, *Leptolyngbya boryana* LEGE 15486 and *Nodosilinea nodulosa* LEGE 06104 water extracts; and G3, involving all the acetone extracts.

According to Figure 3B, G1 is chemically distinct from the other samples due to the PE content; *Cephalothrix lacustris* LEGE 15493, *Leptolyngbya boryana* LEGE 15486, and *Nodosilinea nodulosa* LEGE 06104 water extracts are grouped according to their content in PC, APC, and total proteins (G2), while all the acetone extracts are found in the same group (G3) due to their content in carotenoids, and chlorophyll *a* and its derivatives. The compounds beyond the enzymatic activities are displayed, with the planes formed with the PC1 positive axis (Figure 3B). It can be observed that samples with higher contents in chlorophyll *a* and carotenoids are closely related with elastase and tyrosinase inhibition. This is demonstrated by the strong positive correlation between carotenoids and elastase (0.725, *p* < 0.01) and tyrosinase (0.782, *p* < 0.01) inhibition, with similar observations between chlorophyll *a* and the same enzymes (0.702, *p* < 0.01 and 0.519, *p* < 0.05, respectively). On the other hand, hyaluronidase inhibition is more correlated with PE content (Figure 3B), with a significant positive correlation between the values (0.784, *p* < 0.01). The compounds responsible for the radical scavenging activity of the extracts are displayed, with the planes formed with the PC1 negative axis. The closest correlation is observed for TPC (0.567, *p* < 0.05); other compounds such as total proteins, PC, and APC also contribute to the activity, although to a lower extent (Figure 3B). Regarding the SPF, there is a close correlation between the activity of the extracts and their PBP content, mainly APC (0.835, *p* < 0.01) and PC (0.838, *p* < 0.01), which points to these compounds as predominantly responsible for blocking UV-B radiation. The total protein content also contributed to this biological activity (0.670, *p* < 0.01), with the lowest contribution being observed for PE (0.210, *p* > 0.05). In general, the PCA analysis allowed the grouping of extracts according to the biological activities displayed in the field of skin aging, leading to the conclusion that acetone extracts are more effective in inhibiting enzymes responsible for the degradation of the dermal matrix and loss of skin structure, while aqueous extracts are more effective in scavenging free radicals and protecting skin from the deleterious effects of UV radiation.

As far as we know, this is the first time that a relationship between the chemical composition and biological activities of extracts from these cyanobacteria strains has been established.

## 3. Materials and Methods

### 3.1. Cyanobacteria Biomass Production

Four filamentous cyanobacterial strains, *Cephalothrix lacustris* LEGE 15493 and *Leptolyngbya boryana* LEGE 15486, from Brazilian freshwater ecosystems, and *Leptolyngbya* cf. *ectocarpi* LEGE 11479 and *Nodosilinea nodulosa* LEGE 06104, from Portuguese marine ecosystems, were used in this study. The strains were maintained in the Blue Biotechnology and Ecotoxicology Culture Collection (LEGE CC) at the Interdisciplinary Center of Marine and Environmental Research (CIIMAR). For biomass production purposes, a scale-up culture scheme was set, starting with 40 mL under laboratory-controlled conditions, sequentially scaled up to 4 L. The strains were grown in Z8 medium [53], supplemented with 10 µg/L vitamin B12 and 25 g/L NaCl for marine strains. Cultures were maintained at 25 °C, with a light intensity of 10 μmol photons m^−2^ s^−1^, and with a photoperiod of 14 h light:10 h dark. The fresh biomass was collected after 120 or 150 days of growth (depending on the strain) via filtration, and frozen, freeze-dried, and stored at −20 °C until extract preparation.

### 3.2. Extracts Preparation

Two different extracts were sequentially prepared from each strain: acetone and aqueous. First, the acetone extract was prepared, using 2 g of dry biomass. The biomass was suspended in acetone and extracted for 10 min in an ultrasonic bath (Fisherbrand^®^-FB15053, Loughborough, UK). After the acetone extraction, the resulting pellet was left to dry in the fume hood, and then extracted with 70 mL of distilled water, following the same procedure. Cell debris was removed by centrifugation at 10,000× *g* Gs during 5 min at 4 °C, in a HERAUS Megafuge^TM^ 16R microcentrifuge (Thermo Scientific^TM^, Waltham, MA, USA). Supernatants from each extraction were evaporated under reduced pressure (BUCHI R-210 Rotary Evaporator) (acetone extract), or frozen and lyophilized (aqueous extract). Extraction with the corresponding supernatant was repeated 3 times. The dry extracts were kept at −20 °C until further chemical and biological analysis.

### 3.3. Cell Assays

#### 3.3.1. Cell Culture

Human keratinocytes HaCAT (ATCC), mouse fibroblasts 3T3L1 (ATCC), and human endothelial cells hCMEC (provided by Dr. P. O. Couraud (INSERM)) were used for cytotoxicity evaluation. Cell lines were cultured in Dulbecco’s Modified Eagle Medium (DMEM GlutaMAX™, Gibco, Glasgow, UK), supplemented with 10% (*v/v*) fetal bovine serum (Gibco), 1% penicillin-streptomycin (Pen-Strep 100 IU/mL and 10 mg/mL, respectively) (Gibco) and 0.1% Amphotericin B (Gibco). Cell maintenance and assays were performed at 37 °C in a 5% CO_2_ humidified atmosphere, and the culture medium was renewed every two days. At 80–90% cell confluence, adherent cells were washed with phosphate-buffered saline (PBS, Gibco), detached with TrypLEX express enzyme (1×) (Gibco), passed for maintenance, and seeded for the planned assays.

#### 3.3.2. Cytotoxicity—MTT Assay

Cellular viability was evaluated by the reduction in the 3-(4,5-dimethylthiazole-2-yl)-2,5-diphenyltetrazolium bromide (MTT, Sigma-Aldrich, Germany) assay, as previously reported [26]. All cell lines (endothelial cells, fibroblasts, and keratinocytes) were seeded in 96-well plates, at a density of 1.0 × 10^5^ cells/mL, 3.3 × 10^4^ cells/mL, and 2.5 × 10^4^ cells/mL, respectively. After 24 h of cell adhesion, the culture medium was removed, and the cells were exposed for 24 and 48 h to fresh medium containing the different cyanobacteria extracts in five serial concentrations, from 12.5 to 200 µg/mL. For the acetone extracts, stock solutions were prepared in dimethyl sulfoxide (DMSO) (Gibco), and diluted with DMEM prior to cells’ exposure so that the maximum DMSO concentration did not exceed 1%. Aqueous extracts were prepared in PBS and diluted with DMEM prior to cells’ exposure. The negative control was PBS, and the control for cell death was DMSO 20%. After incubation, the MTT cytotoxicity assay was performed. Briefly, 20 µL of MTT solution was added to each well and incubated at 37 °C for 3 h. Following incubation, the medium was carefully removed, and the purple-colored formazan salts were dissolved in DMSO. The absorbance was read at 550 nm with a Synergy HT multi-detection microplate reader (Biorek, Bad Friedrichshall, Germany) operated by GEN5^TM^ software. The assay was run in quadruplicate and averaged. For reproducibility, each assay was independently repeated at least three times. Cytotoxicity was expressed as the percentage of cell viability, considering 100% viability in the solvent control.

### 3.4. Chemical Profiling of Cyanobacteria Extracts

#### 3.4.1. Total Phenolic Content (TPC)

To determine the TPC of the extracts, a colorimetric assay was used, based on the Folin–Ciocalteu methodology [54] with slight modifications. The acetone extracts were solubilized in DMSO, and the aqueous extracts in water. Briefly, 25 µL of each extract (10 mg/mL) was completely mixed with 25 µL of Folin–Ciocalteu reagent (Sigma-Aldrich), 200 µL of Na_2_CO_3_ solution, and 500 µL of deionized water. For the blank, the Folin–Ciocalteu reagent was replaced by deionized water. The absorbance of the colored product formed was measured at 725 nm, using a Synergy HT Multi-detection microplate reader (Biotek, Bad Friedrichshall, Germany) operated via GEN5^TM^ software. Standard curves for TPC quantification were obtained using seven concentrations of gallic acid (GA) (0.025 to 0.5 mg/mL), prepared in the same solvent as the extracts to be tested (*y* = 2.097*x* + 0.01560, R^2^ = 0.9989, for acetone, and *y* = 2.204*x* + 0.01401, R^2^ = 0.9982, for water, where “*y*” refers to the absorbance and “*x*” refers to the concentration). Three independent determinations were carried out in duplicate. Total phenolic content was expressed as µg GA equivalents (GAE) per mg of dry extract.

#### 3.4.2. Total Proteins

The total protein concentration was determined using the BCA protein assay kit (#23227, Thermo-Scientific). Aqueous extracts were prepared in water, while acetone extracts were prepared in DMSO. Briefly, in a 96-well plate, 25 µL of each extract (1 mg/mL) was mixed with 200 µL of the working reagent. The absorbance was measured at 562 nm, using a Synergy HT Multi-detection microplate reader (Biotek, Bad Friedrichshall, Germany) operated via GEN5^TM^ software. A standard curve (*y* = –126.87*x*^3^ + 547.73*x*^2^ + 483.85*x* − 10.017; R^2^ = 0.999, and *y* = 162.87*x*^3^ − 248.51*x*^2^ + 932.13*x* − 11.715; R^2^ = 0.999, where “*y*” refers to the absorbance and “*x*” refers to the concentration) was obtained for each extract, using nine concentrations of albumin (BSA) (25 to 2000 µg/mL) to quantify the proteins. Three independent experiments were carried out in triplicate. The total protein content was expressed as µg of bovine serum albumin (BSA) equivalents per mg of dry extract.

#### 3.4.3. Pigments

The pigments present in both aqueous and acetone extracts were quantified spectrophotometrically. Chlorophyll *a* and its derivatives were quantified using a calibration curve obtained with the commercial standard (Sigma-Aldrich): *y* = 8.0791*x* − 0.0022 (R^2^ = 0.996), where “*y*” refers to the absorbance and “*x*” refers to the concentration. Total carotenoids were quantified as β-carotene (Sigma-Aldrich), through its calibration curve (*y* = 17.133*x* + 0.0099; R^2^ = 0.990, where “*y*” refers to the absorbance and “*x*” refers to the concentration). Total carotenoid concentration was expressed as µg of β-carotene per mg of dry extract. Calibration curves for both standards were obtained using five different concentrations (0.001 to 0.025 mg/mL). The spectrophotometric determinations were performed in 96-well plates, at 450 nm for β-carotene and 663 nm for chlorophyll *a*, using a Synergy HT Multi-detection microplate reader (Biotek, Bad Friedrichshall, Germany) operated via GEN5^TM^ software.

PBP content was determined spectrophotometrically by measuring the absorbances of the aqueous extracts at different wavelengths (562, 615, and 645 nm), and applying the corresponding formulas, as previously described by Pagels et al. [34]:$$Phycocyanin\left(PC\right)=\frac{A_{615nm}-0.474\times A_{652nm}}{5.34}$$
$$Allophycocyanin\left(APC\right)=\frac{A_{652nm}-0.208\times A_{615nm}}{5.09}$$
$$Phycoerythrin\left(PE\right)=\frac{A_{562nm}-2.41\times PC-0.849\times APC}{9.62}$$

Aqueous extracts were resuspended to a final concentration of 0.5 mg/mL. The experiment was carried out in triplicate, and the results were expressed in µg/mg of dry extract.

### 3.5. Biological Activities

#### 3.5.1. Superoxide Anion Radical (O_2_^•−^) Scavenging

The free radical scavenging assay of O_2_^•−^ was performed to evaluate the antioxidant potential of the cyanobacteria extracts, according to Barbosa and co-workers [55], with minor modifications. The aqueous extracts were prepared in water, while acetone extracts were prepared in DMSO. Five serial dilutions were prepared for each extract and tested in order to evaluate the extracts’ behavior and IC values. All reagents were dissolved in phosphate buffer (19 µM, pH 7.4). A volume of 50 µL of each dilution was mixed with 50 µL of 166 µM β-nicotinamide adenine dinucleotide reduced form (NADH) solution and 150 µL of 43 µM nitrotetrazolium blue chloride (NBT) in a 96-well plate. After the addition of 50 µL of 2.7 µM phenazine methosulphate (PMS), the radical scavenging activity of the samples was monitored with a Synergy HT Multi-detection microplate reader (Biotek, Bad Friedrichshall, Germany) operated via GEN5^TM^ software, in kinetic function, at room temperature for 2 min at 562 nm. Three independent assays were performed in triplicate. GA was used as a positive control. The results were expressed as the percentage of radical scavenging in comparison to the untreated control. The results for the calculated IC values were expressed as the mean ± SD (µg/mL) of at least three independent assays performed in duplicate. The IC values and corresponding dose-response curves were calculated with Graphpad Prism^®^ software (San Diego, CA, USA; version 9, for MacOS).

#### 3.5.2. Hyaluronidase Inhibition

The hyaluronidase inhibition assay was slightly modified from that proposed by Ferreres et al. [56]. Briefly, 25 µL of each extract (9 mg/mL), 175 µL hyaluronic acid (HA) (0.7 mg/mL), and 25 µL of hyaluronidase (HAase) (900 U/mL in NaCl 0.9%) were mixed in a reaction tube. Aqueous extracts were prepared in water, and acetone extracts were prepared in DMSO. After 30 min of incubation at 37 °C, the reaction was stopped by the addition of 25 µL of di-sodium tetraborate (0.8 M in water), followed by incubation for 3 min at 90 °C in a water bath. The reaction tubes were cooled to room temperature before 375 µL of DMAB [4-(Dimethylamino)benzaldehyde] solution was added. After 20 min of incubation at 37 °C, the absorbance of the colored product formed was measured at 560 nm, in a Synergy HT Multi-detection microplate reader (Biotek, Bad Friedrichshall, Germany) operated via GEN5^TM^ software. The negative control was performed in the absence of extract. Disodium cromoglycate (DSCG) was used as a positive control.

Three independent assays were performed in triplicate, and the results were expressed as the percentage of enzyme inhibition in comparison to the untreated control.

#### 3.5.3. Elastase Inhibition

Porcine pancreatic elastase inhibition assay was performed according to Mota and co-workers [57] with slight modifications. Aqueous extracts were prepared in water, while acetone extracts were prepared in DMSO. Briefly, in a 96-well plate, 50 µL of extract was mixed with 90 µL of HEPES buffer (0.1 M), 10 µL of N-succinyl-Ala-Ala-Ala p-nitroanilide substrate (100 µM), 70 µL of acetate buffer (200 mM), and 30 µL of elastase (1 U/mL). The plate was incubated at 37 °C for 10 min, and the absorbance of the reaction product was measured at 405 nm, in a Synergy HT Multi-detection microplate reader (Biotek, Bad Friedrichshall, Germany) operated via GEN5^TM^. The negative control was performed in the absence of extract, and ascorbic acid was used as a positive control. Three independent assays were performed in triplicate. The results were expressed as the percentage of enzyme inhibition in comparison to the untreated control.

#### 3.5.4. Tyrosinase Inhibition

The tyrosinase inhibition assay was performed according to Adhikari et al. [58] with slight modifications. Briefly, in a 96-well plate, 20 µL of each extract was mixed with 100 µL of tyrosinase (30 U/mL in phosphate buffer). Aqueous extracts were prepared in water, while acetone extracts were prepared in DMSO. The mixture was incubated at 30 °C for 8 min. Then, 80 µL of L-DOPA (L-3,4-dihydroxyphenylalanine) solution (2.5 mM in phosphate buffer) was added, and the absorption (T0, absorbance at time “zero”) was immediately measured with a Synergy HT Multi-detection microplate reader (Biotek, Bad Friedrichshall, Germany) operated via GEN5^TM^ software, at 475 nm. The determination of the absorbance at T0 (before the reaction product was formed) allowed the elimination of possible interferences due to the natural color of the extracts under study. After 8 min of incubation at 30 °C, the absorbance was measured again (T8). The percentage of tyrosinase inhibition in the presence of cyanobacteria extracts was calculated in comparison to the untreated (negative) control, where the difference between the absorbances (T8–T0) corresponds to 100% of enzyme activity. The negative control was performed in the absence of extract, and kojic acid was used as a positive control. Three independent assays were performed in triplicate. The results were expressed as the percentage of enzyme inhibition in comparison to the untreated control.

#### 3.5.5. Sun Protection Factor (SPF)

The in vitro sun protector factor was determined according to Rohr and co-workers [59] with slight modifications. Aqueous extracts were prepared in water, while acetone extracts were prepared in acetone. Briefly, the absorbance of 2 mL of each extract (200 and 1000 µg/mL), was measured in a spectrophotometer (from 290 to 320 nm, 5 in 5 nm). The SPF was calculated using the formula proposed by Mansur [60]:$$SPF_{spectrophotometric}=CF\times \overset{320}{\underset{290}{\sum }}EE\left(\lambda \right)\times I\;\left(\lambda \right)\times Abs\left(\lambda \right)$$
where EE(λ) is the erythemal effect spectrum, I (λ) is the solar intensity spectrum, Abs(λ) is the absorbance of extract, and CF is the correction factor (28) determined using a commercial sunscreen with a known SFP value of 30.

### 3.6. Statistical Analysis

Statistical analysis was performed using IBM SPSS statistics software (version 23.0 for MacOS, IBM Corporation, New York, NY, USA, 2015). Data were analyzed for normality and the homogeneity of variances by Kolmogorov–Smirnov and Leven’s tests, then submitted to a one-way ANOVA using a Tukey’s HSD (honest significant difference) as a post-hoc test, or to an unpaired *t*-test. A Pearson correlation test was used to compare the normalized expression data between the chemical profiles and biological activities of cyanobacteria extracts.

### 3.7. Principal Component Analysis (PCA)

PCA was used to transform a number of potentially correlated variables into a number of relatively independent variables, able to be ranked based on their contribution to explaining the variation of the whole data set [61]. The relatively important components of high-dimensional patterns can be successfully identified. The original high-dimensional data can be mapped onto a lower dimensional space, and therefore the complexity of a high-dimensional pattern classification problem is greatly reduced [62]. For the present study, pattern recognition based on PCA was performed using IBM SPSS statistics software (version 23.0 for MacOS, IBM Corporation, New York, NY, USA, 2015). The data matrix consisted on the metabolites present in the aqueous and acetone extracts of the four cyanobacteria strains, and their activity at the highest concentration tested.

## 4. Conclusions

The biological activity displayed by the cyanobacteria extracts analyzed herein proved to be clearly correlated. According to the presence of bioactive PBPs, the aqueous extracts were the most effective for UV protection which, together with their radical scavenging capacity, suggests them as promising ingredients to be used in anti-aging formulations aimed at preventing skin aging exacerbated by external factors. On the other hand, the acetone extracts proved more effective in inhibiting the activity of enzymes responsible for the degradation of the dermal matrix and loss of skin structure, thus being more suitable for age-related skin aging. The sequential extraction scheme we propose could prove advantageous, allowing the obtainment of chemically different bioactive extracts through the monetization of cyanobacteria biomass, making the process more sustainable and economically attractive. Altogether, cyanobacteria extracts proved worthy of further exploitation in the field of skin aging, targeting the search for natural, safe, and sustainable ingredients for cosmetic formulations.

## Figures and Tables

**Figure 1 marinedrugs-20-00183-f001:**
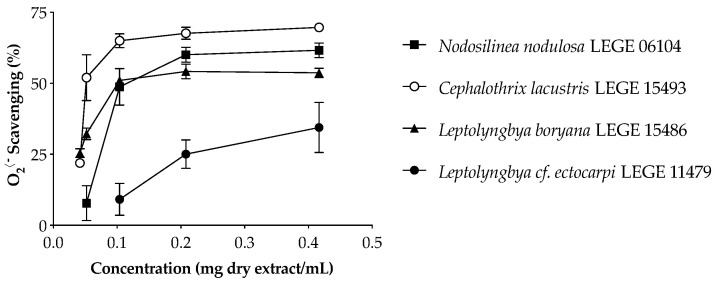
Superoxide anion radical (O_2_^•−^) scavenging activity of cyanobacteria aqueous extracts. Values are expressed as the mean ± SD of three determinations.

**Figure 2 marinedrugs-20-00183-f002:**
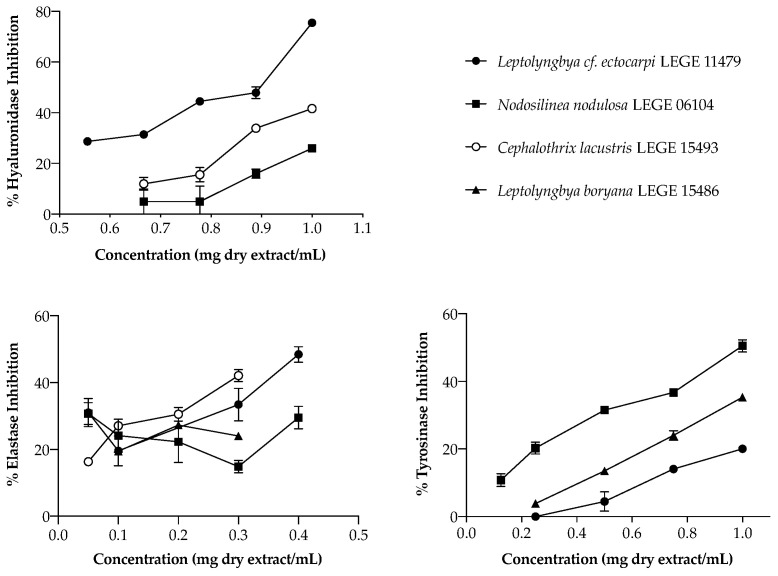
Inhibitory activity of cyanobacteria acetone extracts over the enzymes hyaluronidase, elastase, and tyrosinase. Values are expressed as the mean ± SD of three determinations. The behavior of *Leptolyngbya* cf. *ectocarpi* LEGE 11479 on hyaluronidase corresponds to the aqueous extract.

**Figure 3 marinedrugs-20-00183-f003:**
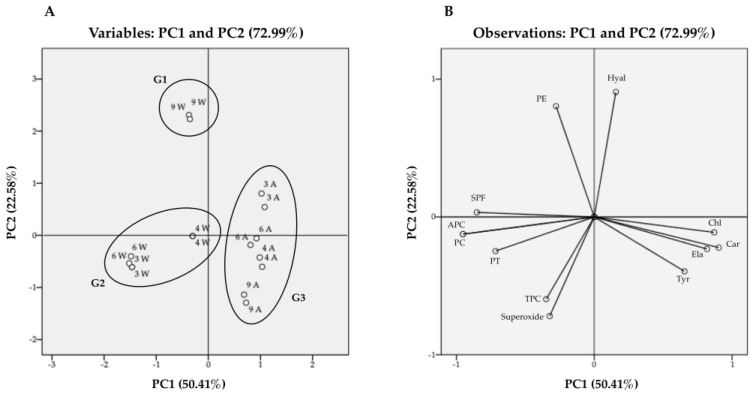
Projection of cyanobacteria extracts (**A**) [variables: *Nodosilinea nodulosa* LEGE 06104 (LEGE 06104W and LEGE 06104A), *Leptolyngbya* cf. *ectocarpi* LEGE 11479 (LEGE 11479W and LEGE 11479A), *Cephalothrix lacustris* LEGE 15493 (LEGE 15493W and LEGE 15493A), and *Leptolyngbya boryana* LEGE 15486 (LEGE 15486W and LEGE 15486A), where “W” refer to aqueous extracts and “A” to acetone extracts] and loadings (**B**) by chemical composition and bioactivities [variables: total phenolic content (TPC), total proteins (PT), carotenoids (Car), chlorophyll a (Chl), radical scavenging activity (Superoxide), hyaluronidase (Hyal), elastase (Ela), tyrosinase (Tyr), phycocyanin (PC), phycoerythrin (PE), allophycocyanin (APC), SPF at 1000 µg/mL (SPF)] into the plane composed by the principal components PC1 and PC2 containing 72.99% of the total variance.

**Table 1 marinedrugs-20-00183-t001:** Cyanobacteria extraction yield (% *w*/*w*).

Strains	Solvent	
	Acetone	Water
*Nodosilinea nodulosa* LEGE 06104	0.6	9.0
*Leptolyngbya* cf. *ectocarpi* LEGE 11479	1.6	14.5
*Cephalothrix lacustris* LEGE 15493	2.0	9.5
*Leptolyngbya boryana* LEGE 15486	2.9	22.0

**Table 2 marinedrugs-20-00183-t002:** Total phenolic content (µg GAE/mg _dry extract_) of cyanobacteria extracts ^1,2,3^.

Strains	Solvent	
	Acetone	Water
*Nodosilinea nodulosa* LEGE 06104	11.23 ± 1.55	6.52 ± 0.38
*Leptolyngbya* cf. *ectocarpi* LEGE 11479	17.59 ± 2.29	8.49 ± 1.18
*Cephalothrix lacustris* LEGE 15493	nd	13.75 ± 0.28
*Leptolyngbya boryana* LEGE 15486	7.10 ± 1.73	13.98 ± 0.90

^1^ GAE, gallic acid equivalents; ^2^ Mean ± SD of three independent experiments; ^3^ nd, not detected.

**Table 3 marinedrugs-20-00183-t003:** Total protein content (µg BSA/mg _dry extract_) in cyanobacteria extracts ^1^.

Strains	Solvent	
	Acetone	Water
*Nodosilinea nodulosa* LEGE 06104	134.76 ± 2.41	169.18 ± 2.21
*Leptolyngbya* cf. *ectocarpi* LEGE 11479	136.63 ± 3.60	185.69 ± 0.78
*Cephalothrix lacustris* LEGE 15493	218.23 ± 3.48	521.18 ± 0.60
*Leptolyngbya boryana* LEGE 15486	177.27 ± 5.89	314.60 ± 5.90

^1^ Values are expressed as the mean ± SD of three determinations.

**Table 4 marinedrugs-20-00183-t004:** Phycobiliprotein content (µg/mg _dry extract_) in cyanobacteria aqueous extracts ^1^.

Strain	Phycobiliprotein		
	Phycocyanin	Allophycocyanin	Phycoerythrin
*Nodosilinea nodulosa* LEGE 06104	50.11 ± 0.26	10.14 ± 0.60	6.20 ± 0.08
*Leptolyngbya* cf. *ectocarpi* LEGE 11479	53.94 ± 0.24	8.77 ± 1.05	138.73 ± 0.33
*Cephalothrix lacustris* LEGE 15493	115.03 ± 0.40	27.85 ± 0.16	18.93 ± 0.03
*Leptolyngbya boryana* LEGE 15486	154.07 ± 0.26	46.43 ± 0.06	7.51 ± 0.09

^1^ Values are expressed as the mean ± SD of two determinations.

**Table 5 marinedrugs-20-00183-t005:** Carotenoid and chlorophyll *a* content (µg/mg _dry extract_) in the cyanobacteria acetone extracts ^1^.

Strains	Compounds	
	Carotenoids	Chlorophyll *a*
*Nodosilinea nodulosa* LEGE 06104	125.34 ± 5.0	117.89 ± 4.86
*Leptolyngbya* cf. *ectocarpi* LEGE 11479	89.07 ± 5.0	114.25 ± 2.73
*Cephalothrix lacustris* LEGE 15493	137.50 ± 2.8	264.94 ± 11.84
*Leptolyngbya boryana* LEGE 15486	159.39 ± 5.0	190.02 ± 6.38

^1^ Values are expressed as the mean ± SD of three determinations.

**Table 6 marinedrugs-20-00183-t006:** Inhibitory concentration (IC) values (µg _dry extract_/mL) of cyanobacteria extracts obtained for superoxide anion radical scavenging ^1^.

Strains		Solvent			
		Acetone		Water	
		IC_25_	IC_50_	IC_25_	IC_50_
*Nodosilinea nodulosa* LEGE 06104		1121.50 ± 89.80	-	73.0 ± 4.0	101.3 ± 22.9
*Leptolyngbya* cf. *ectocarpi* LEGE 11479		580.0 ± 29.7	1190.5 ± 108.2	42.5 ± 0.5	-
*Cephalothrix lacustris* LEGE 15493		1080.0 ± 254.9	-	43.0 ± 0.0	65.5 ± 7.8
*Leptolyngbya boryana* LEGE 15486		1190.3 ± 110.8	-	198.7 ± 23.3	99.5 ± 0.7

^1^ Values are expressed as the mean ± SD of three determinations.

**Table 7 marinedrugs-20-00183-t007:** Sun protection factor of cyanobacteria extracts ^1^.

Strains	Solvent			
	Acetone		Water	
	200 µg/mL	1000 µg/mL	200 µg/mL	1000 µg/mL
*Nodosilinea nodulosa* LEGE 06104	3.50 ± 0.07	6.30 ± 0.09	1.30 ± 0.02	7.27 ± 0.05
*Leptolyngbya* cf. *ectocarpi* LEGE 11479	10.74 ± 0.41	7.44 ± 0.15	1.97 ± 0.02	11.54 ± 0.00
*Cephalothrix lacustris* LEGE 15493	7.79 ± 0.51	5.06 ± 0.16	2.87 ± 0.02	14.86 ± 0.42
*Leptolyngbya boryana* LEGE 15486	19.22 ± 1.50	12.00 ± 0.78	3.05 ± 0.16	17.16 ± 0.02

^1^ Values are expressed as the mean ± SD of three determinations.

## Data Availability

Not applicable.

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
