# Peer review of "Cyanobacteria Secondary Metabolites as Biotechnological Ingredients in Natural Anti-Aging Cosmetics: Potential to Overcome Hyperpigmentation, Loss of Skin Density and UV Radiation-Deleterious Effects"

_marinedrugs, 2022, doi:10.3390/md20030183_

Round 1

Reviewer 1 Report

Dear authors,

I found your manuscript quite interesting and might main concerns come fro your comparisons with previous published work. I would mentioned previous published articles but I will avoid comparisons. Cyanobacteria can change their pigmentation and metabolic rates depending of many factors: -light conditions -Nutrients -Time of the day -cell density -etc.

I think your study is value as far as people reproduce your conditions, there fore I would include details such as optical density/time of the circadian cycle when they were harvested.

I think there is a high potential on the use of cyanobacteria in cosmetic.

Author Response

ANSWER TO REVIEWER 1

First we would like to thank the reviewer for his appreciation of our work, which contributed to improve its quality and clarification. We have addressed his questions and made the requested modifications in the revised manuscript. Please find bellow our answers point by point.

REVIEW FOR THE ARTICLE ENTITLED

Dear authors,

I found your manuscript quite interesting and might main concerns come from your comparisons with previous published work. I would mentioned previous published articles but I will avoid comparisons. Cyanobacteria can change their pigmentation and metabolic rates depending of many factors: -light conditions -Nutrients -Time of the day -cell density -etc.

We understand the reviewers’ comment. In fact, cyanobacteria metabolism suffer a significant variation depending on abiotic factors, of which photoperiod has a great impact, especially regarding pigments production. We have revised our manuscript accordingly, in order to clarify comparisons with other works, where the cultivation conditions are different. Nevertheless, we have maintained the comparisons of the results obtained in the present work with others previously reported by our research group, since the conditions of biomass cultivation were similar. The modifications have been highlighted in the revised version of our manuscript.

I think your study is value as far as people reproduce your conditions, therefore I would include details such as optical density/time of the circadian cycle when they were harvested.

We understand and agree with the reviewers’ comment. Taking into account the influence of abiotic factors in the metabolome of cyanobacteria, it is of great importance that their cultivation conditions can be mimic in order to obtain similar results. As suggested by the reviewer, a better description of cyanobacteria cultivation and harvesting conditions was added to the methodology section. Please see the revised version of our manuscript.

I think there is a high potential on the use of cyanobacteria in cosmetic.

We thank the reviewer for his appreciation of this valuable resource.

Author Response

Please find our answers in the PDF file in attachment.

Round 2

Reviewer 2 Report

I have seen the revised form, and I suggest accepting the article in its present form.